# Support Vector Machine–Based Model for 2.5–5.2 GHz CMOS Power Amplifier

**DOI:** 10.3390/mi13071012

**Published:** 2022-06-27

**Authors:** Shaohua Zhou, Cheng Yang, Jian Wang

**Affiliations:** 1School of Microelectronics, Tianjin University, Tianjin 300072, China; zhoushaohua@tju.edu.cn (S.Z.); ych2041@tju.edu.cn (C.Y.); 2Qingdao Institute for Ocean Technology, Tianjin University, Qingdao 266200, China

**Keywords:** PA, temperature, SVM, model, complementary metal oxide semiconductor (CMOS)

## Abstract

A power amplifier (PA) is the core module of the wireless communication system. The change of its specification directly affects the system’s performance and may even lead to system failure. Furthermore, change in the PA specification is closely related to changes in temperature. To study the influence of PA specification change on the system, we used a support vector machine (SVM) to model the temperature characteristics of PA. For SVM modeling, the question of how much experimental data should be used for modeling to meet the requirements is a constant problem. To address this issue, we investigate the effect of different amounts of training data on the modeling of SVM models. The results show that only 75% of the experimental data needs to be used in the modeling process to satisfy the modeling requirements of the SVM model. The number of measurement points required in the PA specification degradation experiment can be reduced by 25%. The results of this paper serve as a guide for planning the number of experimental measurement points and reducing the measurement cost and measurement time.

## 1. Introduction

Power amplifiers (PAs) are the core components of wireless transceivers and have been the focus of research in wireless communications [1,2]. In addition, power amplifiers are a critical element of wireless transmitters and affect modern wireless communication systems and next-generation radars [3,4].

Due to heating and cooling effects, temperature changes can degrade the PA’s specifications [5,6]. For example, when PAs are used in wireless transmitter arrays (e.g., multiple-input multiple-output (MIMO) or radar arrays) [5], the temperature of the PA is not only determined by its self-heating [5,7,8] but is also affected by thermal coupling with nearby PAs [9], leading to degradation of its specifications. In severe cases, this will lead to the failure of the wireless communication system in which it is located [10]. Therefore, to know the relationship between the failure of the wireless communication system and the degradation of the PA specification, it is necessary to investigate the relationship between the degradation of PA specifications and temperature variations. Furthermore, to know the relationship between the specification of the PA and the temperature change, it is necessary to model the evolution of the specification of the PA with temperature.

Support vector machine (SVM) is a new machine-learning algorithm proposed by V. Vapnik et al. in the 1990s [11,12,13,14]. It minimizes the actual risk by seeking structural risk minimization and pursues the optimal result under the limited information condition [13]. There are two main applications of support vector machines, one is classification, and the other is regression [15]. Moreover, the modeling of power amplifiers belongs to nonlinear regression. Therefore, the emergence of support vector machines opens a new way to solve the nonlinear relationship of power amplifiers [16,17,18].

In recent years, SVM has been successfully and widely used for the behavior modeling and optimization of PAs. J. B. King et al. proposed a novel technique for modeling the dynamic behavior of PAs based on a time-delay support vector regression (SVR) approach [19]. The proposed model accurately predicts PA behavior under different input powers and provides better performance with reasonable complexity. L. Qi et al. proposed a novel behavioral model of PA based on multi-segment SVR [20]. In this model, the input signal is divided into two parts according to the amplitude of the input signal, and each part is modeled separately using SVR. The proposed model can effectively reduce parameter extraction and signal processing time. Lin Maoliu et al. used an optical modeling method based on SVM to describe the nonlinear characteristics of radio-frequency (RF) PA: the nonlinear scattering function [21]. This function allows modeling of the accurate behavior of the RF PA under large signals. Finally, T. J. Brazil et al. proposed an automatic optimization method for matching networks using SVR to control a continuous class-F PA [22]. The proposed method not only significantly surpasses the genetic optimizer and simplified real frequency technique (SRFT) of the advanced design system (ADS) but also achieves similar performance to Gaussian process regression (GPR). To realize the system-level simulation of a PA link affected by a broadband modulation signal, J. Cai et al. proposed a new behavior model of PA based on SVR [23]. This model can achieve more accurate predictions than the traditional double-input (DIDO) model based on the memory polynomial (MP).

This paper uses SVM to model the temperature characteristics of a 2.5–5.2 GHz complementary metal oxide semiconductor (CMOS) PA for the first time. In addition, the influence of different amounts of training data on the model is discussed. The results show that, when the accuracy order of the model is the same, the model’s training data are reduced by 25%, and the training time of the model can be shortened by about 2.1 times.

The organizational structure of this paper is as follows. The second and third parts briefly introduce the PA’s measurement and modeling process. The fourth part focuses on the discussion and analysis of the modeling results. Finally, the full text is summarized, and the direction and content of the follow-up work are pointed out.

## 2. Measurement

We measured temperature characteristics on a 2.5–5.2 GHz CMOS Class-A PA and modeled its temperature characteristics using SVM. The schematic of the PA is shown in Figure 1. The PA consists of a two-stage amplifier. Inside the dashed box are on-chip components, and outside the dashed box are off-chip components. Each stage amplifier is implemented with a stacked structure of three transistors, and a better broadband characteristic can be achieved by using a stacked structure.

The PA used in the experiment was fabricated using a 0.18 µm CMOS 1P6M process, and the PA was fixed to the printed circuit board (PCB) with FR-4 as the substrate by gold wire bonding to facilitate the measurement, as shown in Figure 2a.

The measurement connection diagram and this experiment’s physical connection diagram are shown in Figure 2b,c, respectively. From Figure 2b, the temperature characteristic measurement of S-parameters of the PA mainly includes a vector network analyzer (VNA), a DC power supply, an environmental chamber, and high- and low-temperature cables and power cables. The object to be measured is placed in the environmental chamber (seen in Figure 2c) and connected to the VNA and DC power supply through the high- and low-temperature cables and power supply cables, respectively.

The environmental chamber (Vötsch Industrietechnik SC^3^ 1000 MHG) provides a high- and low-temperature experimental environment for the PA from −40 to 125 °C. The S-parameters of the PA are measured by a VNA (R&S ZVA24). The DC power supply (R&S HMP4040) provides the required DC bias for the PA.

## 3. Modeling Process

SVM is a supervised learning model widely used in statistical classification and regression analysis. A nonlinear mapping algorithm for linearly indivisible training data can transform the samples into a high-dimensional feature space. Furthermore, it constructs the optimal segmentation hyperplane in the feature space based on the structural risk minimization theory, making the learner achieve global optimization. Therefore, SVM has been widely used for behavior modeling and optimizing PAs [15].

Modeling the specification degradation behavior of an RF power amplifier belongs to the nonlinear regression problem formulated as follows [15]: given the data set {(*x_1_*, *y_1_*), (*x_2_*, *y_2_*), …, (*x_i_*, *y_i_*), …, (*x_l_*, *y_l_*)}, where *x_i_* ∈ *R_n_*, *y_i_* ∈ *R*, *i* = 1, 2, 3, …, *l*, the relationship *y* = *f*(*x*) between the inputs and outputs is obtained to predict the output value y corresponding to any input *x*.

The linear regression problem is considered first, then the nonlinear regression problem is discussed based on the linear regression problem. Finally, the linear fit function can be expressed as [15]
(1)f(x)=w⋅x+b
where *w* is the weight factor and *b* is the threshold value.

Assuming that the given data set can be fitted without error using Equation (1) while satisfying the requirement of accuracy *ε*, we have [15]
(2)$$\{\begin{array}{c} y_{i}-w\cdot x_{i}-b\leq \varepsilon \\ w\cdot x_{i}+b-y_{i}\leq \varepsilon \end{array},$$

Considering the optimization objective of minimization and the effect of unavoidable noise in the modeling process, a relaxation factor is introduced in Equation (2).
(3)$$\{\begin{array}{c} \xi_{i}\geq 0\\ \xi_{i}^{*}\geq 0 \end{array},$$

Then the optimization problem in Equation (2) can be written as [15]
(4)$$\{\begin{array}{c} \min=\frac{1}{2}{\left\|w\right\|}^{2}+C\sum_{i=1}^{l}\left(\xi_{i}+\xi_{i}^{*}\right)\\ s.t.=\{\begin{array}{c} y_{i}-w\cdot x_{i}-b\leq \varepsilon +\xi_{i}\\ w\cdot x_{i}+b-y_{i}\leq \varepsilon +\xi_{i}^{*} \end{array} \end{array},$$
where the constant *C* > 0 indicates the degree of penalty for data samples that exceed the allowable error *ε*.

To solve the inequality constraint in the above equation, we introduce the dyadic variables (Lagrange multipliers) into Equation (4), then the dyadic form of the original optimization problem can be expressed as [15]
(5)$$\{\begin{array}{c} \max=W(\alpha ,\alpha^{*})=-\frac{1}{2}\sum_{i,j=1}^{l}(\alpha_{i}-\alpha_{i}^{*})(\alpha_{j}-\alpha_{j}^{*})(x_{i}\cdot x_{j})+\sum_{i=1}^{l}(\alpha_{i}-\alpha_{i}^{*})y_{i}-\varepsilon \sum_{i=1}^{l}\left(\alpha_{i}+\alpha_{i}^{*}\right)\\ s.t.=\{\begin{array}{c} \sum_{i=1}^{l}(\alpha_{i}-\alpha_{i}^{*})=0\\ 0\leq \alpha_{i},\alpha_{i}^{*}\leq C \end{array} \end{array},$$

The linear fit function of the SVM is obtained by solving Equation (5) [15].
(6)$$f(x)=w\cdot x+b=\sum_{i=1}^{l}\left(\alpha_{i}-\alpha_{i}^{*}\right)x_{i}\cdot x+b,$$

For nonlinear regression problems, it is necessary to map the data to a high-dimensional feature space through a nonlinear mapping and then solve the linear regression problem in the high-dimensional feature space. The dimensionality of the high-dimensional feature space is very high, even infinite, so the “kernel function” is introduced [15].
(7)$$K(x_{i},x_{j})=\phi (x_{i})\cdot \phi (x_{j}),$$

The expression of the nonlinear fitting function can be obtained according to Equations (6) and (7) as [15]
(8)$$f(x)=w\cdot x+b=\sum_{i=1}^{l}\left(\alpha_{i}-\alpha_{i}^{*}\right)K(x_{i},x)+b,$$

Figure 3 shows the modeling flow of a 2.5–5.2 GHz CMOS Class A PA’s temperature characteristics based on SVM.

The modeling steps of PA temperature characteristics based on SVM mainly include the following steps:

Step 1: Obtain the data required for modeling from the temperature characteristics of the PA. The data are mainly derived from the S-parameters of PA at different temperatures.

The input variables in the model are frequency and temperature, and the output variables are the four S-parameters of the PA.

Step 2: Divide the data in the first step into two parts, half as training data and half as test data.

Step 3: The training data is used to train the SVM model until the model training is complete.

The model’s algorithm is mainly referred to in the literature [15,24]. Therefore, the model’s training is carried out on the MATLAB software platform. In the model’s training process, we directly call the relevant instructions in the SVM toolbox written by S. R. Gunn of the University of Southampton, UK, in MATLAB to train the SVM model [15].

Step 4: Validate the model with test data and calculate the model’s error. In this model, the maximum acceptable order of magnitude of error is 10^−1^. This is mainly chosen based on the accuracy of the measurement.

Step 5: Compare the magnitude of the model error with the expected value. If the requirements are met, the model training is complete. If the conditions are not met, the model needs to be retrained. Then repeat steps 4 and 5.

## 4. Modeling Results and Discussion

### 4.1. Modeling Results of S11

The modeling results of the temperature characteristics of S11 based on SVM are shown in Figure 4. From the modeling results, the measured results of S11 at −35 and 25 °C better agree with the model. When the temperature is 125 °C, the deviation between the measurement results of S11 and the model is relatively large in the frequency range of 4.6–5.2 GHz. From the curves of S11 at the three temperatures, the shapes of the curves of S11 at the two temperatures of −35 and 25 °C are relatively similar, while the curve of S11 at 125 °C changes exactly in the trend of the frequency range of 4.6–5.2 GHz. This indicates that the model’s error is closely related to the data type.

### 4.2. Modeling Results of S12

Figure 5 shows the modeling results of S12. The modeling results of S12 show that there is a relatively large gap between the model and the actual measurement results, especially when the temperature is −35 °C. From the shape of the curves, the shapes of the curves of S11 at 25 and 125 °C are relatively similar, while the shapes of the curves of S11 at −35 °C are very different. This also shows that the model’s error is closely related to the data type.

### 4.3. Modeling Results of S21

The modeling results of the temperature characteristics of S21 based on SVM are shown in Figure 6. From the results of the model, the actual measurement results of S21 match well with the model. Therefore, it indicates that SVM is effective for modeling the temperature characteristics of S21 of this PA. In addition, from the temperature characteristics of S21, S21 decreases with the increase in temperature, mainly caused by the decrease of carrier mobility with the temperature rise. The following is an analysis of the causes of the reduction of small-signal gain (S21) with increasing temperature.

The transduction in the unsaturated and saturated regions, respectively [25,26,27].
(9)$$g_{m}{}_{L}=\frac{\partial I_{D}}{\partial V_{GS}}=\frac{W\mu_{n}C_{ox}}{L}V_{DS},$$
(10)$$g_{ms}=\frac{\partial I_{D}(\mathrm{sat})}{\partial V_{GS}}=\frac{W\mu_{n}C_{ox}}{L}(V_{GS}-V_{T}),$$
where *W* is the gate width, *μ_n_* is the carrier mobility, *C_ox_* is the gate oxide capacitance per unit area, *L* is the gate length, *V_GS_* is the gate voltage, *V_T_* is the threshold voltage, and *V_DS_* is the drain voltage.

There are two main temperature-dependent factors in Equations (9) and (10), the threshold voltage and the carrier mobility. The threshold voltage has a negative temperature coefficient [27]. Thus, the transconductance increases with increasing temperature, causing the transconductance to have a positive temperature coefficient. The carrier mobility versus temperature can, then, be expressed as [27].
(11)$$\mu_{n}(T)=\mu_{n}(T_{0}){\left(\frac{T}{T_{0}}\right)}^{-3/2},$$
where *T_0_* = 300 K.

The carrier mobility from Equation (11) has a negative temperature coefficient. As the temperature increases, the decrease in carrier mobility reduces transconductivity, i.e., transconductivity has a negative temperature coefficient. Combining the different effects of the above two factors, threshold voltage and carrier mobility, according to the expression of transconductance, we can see that,

(i) When the saturation voltage (*V_GS_*-*V_T_*) is relatively large (i.e., *V_GS_* > > *V_T_*), the effect of the threshold voltage on the transconductance can be neglected. Then, the temperature characteristics of the transconductance depend mainly on the temperature dependence of the carrier mobility, i.e., the transconductance has a negative temperature coefficient.

(ii) When the saturation voltage (*V_GS_*-*V_T_*)) is relatively small (i.e., *V_GS_*~*V_T_*), the temperature characteristics of the transconductance depend mainly on the temperature dependence of the threshold voltage, i.e., the transconductance has a positive temperature coefficient.

In practice, the saturation voltage (*V_GS_*-*V_T_*)) is often chosen to be relatively large to obtain a large transconductance so that the effect of the threshold voltage can be disregarded. In addition, the effect of carrier mobility tends to dominate due to its exponential nature [27]. Therefore, the transconductance usually has a negative temperature coefficient, i.e., it decreases with increasing temperature, and the transconductance is usually considered to be the gain of the transistor [26]. Thus, it can be observed that the small-signal gain of PA degrades with increasing temperature.

### 4.4. Modeling Results of S22

The modeling results of the S22 temperature characteristics are shown in Figure 7. From the modeling results, the agreement between the measured results and the model of S22 is relatively good at 25 and 125 °C. However, when the temperature is −35 °C, the agreement between the model and the measurement results is better in the frequency range of 2.75–4.85 GHz. In the frequency ranges of 2.5–2.75 GHz and 4.85–5.2 GHz, there are some deviations between the model and the actual measurement results.

### 4.5. The Effect of the Amount of Training Data on the Model

In addition to modeling the temperature characteristics of PAs using SVM, the impact of different amounts of training data on the accuracy and training time of the model were compared, and the results are shown in Table 1. More-intuitive results are shown in Figure 8 and Figure 9. From Table 1 and Figure 8 and Figure 9,

(1) When the number of model training data and the temperature are the same, the training time of the model corresponding to the four S parameters, S11, S12, S21, and S22, is the same, but the error varies greatly. For example, when the number of training data is 50%, the test errors of the models corresponding to the four S parameters of S11, S12, S21, and S22 were 8.7065 × 10^−1^, 10.5266, 9.6149 × 10^−2^, and 9.4292, respectively, under the environment of −35 °C. This indicates that the accuracy of the model is close.

(2) When the number of training data is the same among the models corresponding to the four S-parameters S11, S12, S21, and S22, the model accuracy of the temperature characteristic of S21 is the highest. This again shows that the model’s accuracy is closely related to the data type. Because this paper focuses on whether SVM can be used to model a PA’s temperature characteristics, this paper does not carry out in-depth research on the relationship between model accuracy and data type. However, this is a topic worthy of discussion. By classifying data types, we can discuss the relationship between model accuracy and data types and provide a reference for selecting models in the modeling process.

In general, SVM is a better choice for modeling the degenerate behavior of S21 of PA.

Finally, we can know from Table 1 and Figure 10 that reducing the number of training data of the model from 50% to 25% can achieve the model accuracy of the same order of magnitude and shorten the training time of the model by about 2.1 times. Using 50% of the data as training data indicates that using 50% of the data as validation data is sufficient. The same effect as that of using 50% of the data as training data can be achieved by using 25% of the data as training data, indicating that using 25% of the data as training data can satisfy the requirement. It also shows that using 50% of the data as validation data and 25% as training data can meet the modeling requirements of the model. It means that only 75% of the data is needed for modeling. This will significantly reduce the time and cost of measurement and help researchers know the PA’s temperature characteristics faster. However, it should be noted that the model accuracy under the three different amounts of training data has great room for improvement. Since the main concern of this paper is that the SVM can be widely used in the behavior modeling of PAs, it can also be used in the temperature characteristic modeling of PAs. Therefore, there is not much research on the optimization of the model. Fortunately, the results show that an SVM can be successfully used to model the temperature characteristics of PAs.

## 5. Conclusions

SVM has been successfully and widely used in the behavior modeling of PAs and achieved remarkable results. This paper uses a 2.5–5.2 GHz CMOS Class-A PA as an example to measure the temperature characteristics, and the SVM is used to model its temperature characteristics for the first time. The results show that the SVM can be used to model the temperature characteristics of PAs. At the same time, this paper also studies the relationship between the quantity and the training data. The results show that only 75% of the experimental data is needed for modeling while ensuring the same order of magnitude of model accuracy. That is, the number of test points can be reduced by 25% in developing a measurement scheme.

The follow-up work of this paper can focus on the relationship between model accuracy and data type and how to optimize the model to improve model accuracy. In particular, the relationship between model accuracy and data type should be studied. The relationship between different models and data types should be established by classifying and summarizing data types. This will provide valuable reference and guidance for selecting models in the modeling process, which is of great significance and value. In addition, other models can be considered for modeling PA specification degradation behavior in the subsequent research process, and a comparison between the modeling results of SVM and other models can be made to provide a reference for the selection of the model for PA specification degradation behavior.

## Figures and Tables

**Figure 1 micromachines-13-01012-f001:**
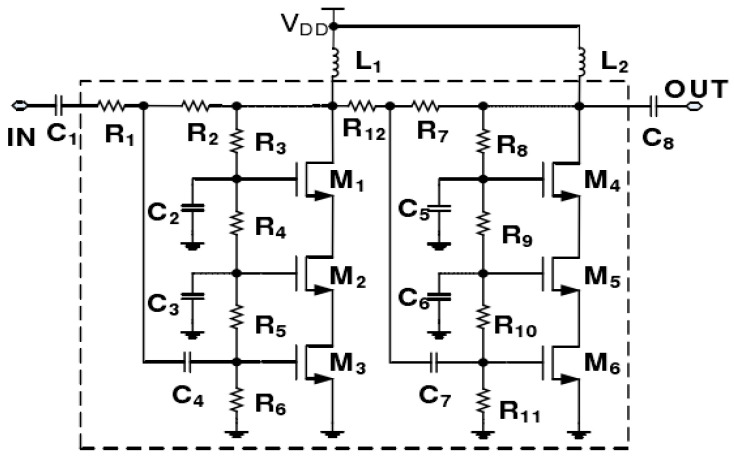
The schematic of the 2.5–5.2 GHz CMOS Class-A PA.

**Figure 2 micromachines-13-01012-f002:**
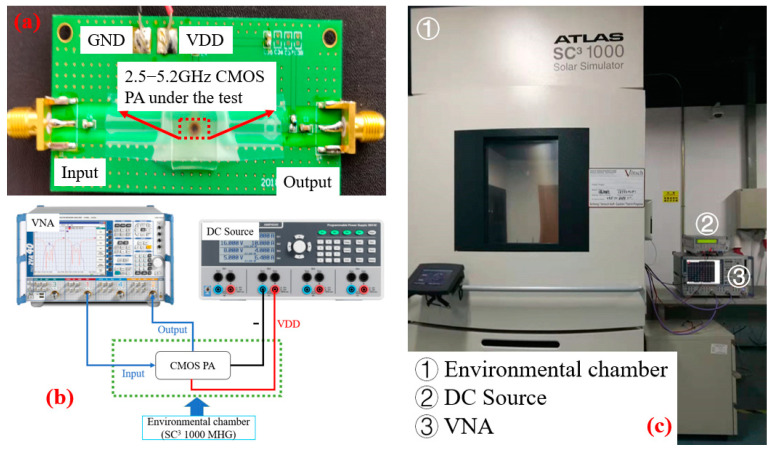
The photograph of the (**a**) object to be measured; (**b**) connection diagram of the measurement; (**c**) physical diagram of the measurement.

**Figure 3 micromachines-13-01012-f003:**
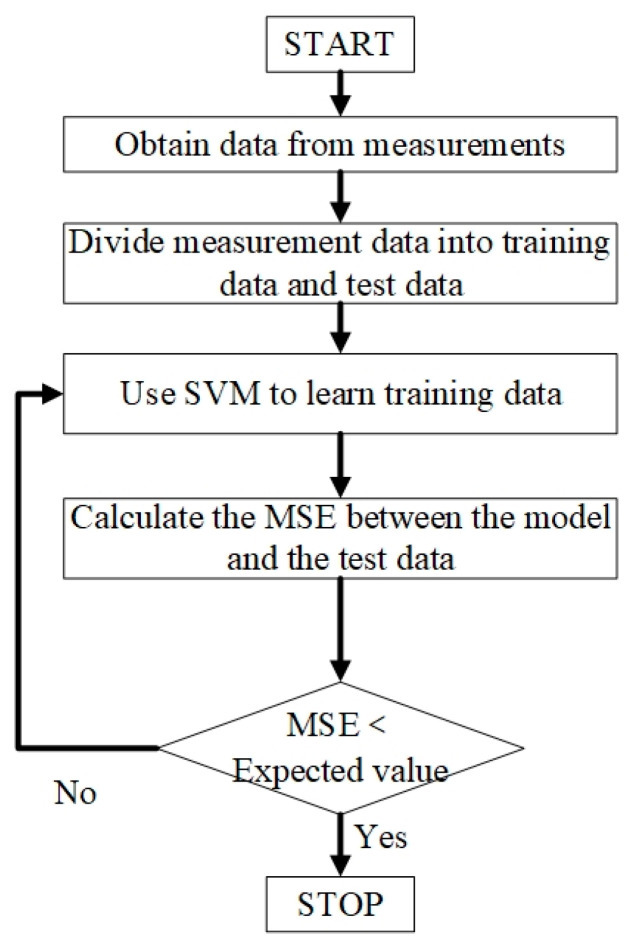
Modeling flow chart of PA temperature characteristics based on SVM.

**Figure 4 micromachines-13-01012-f004:**
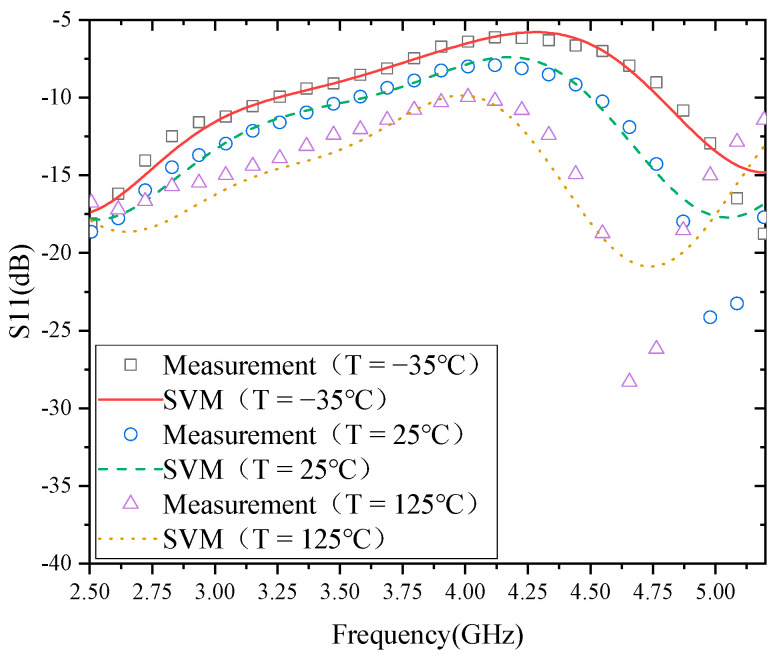
Modeling results of S11.

**Figure 5 micromachines-13-01012-f005:**
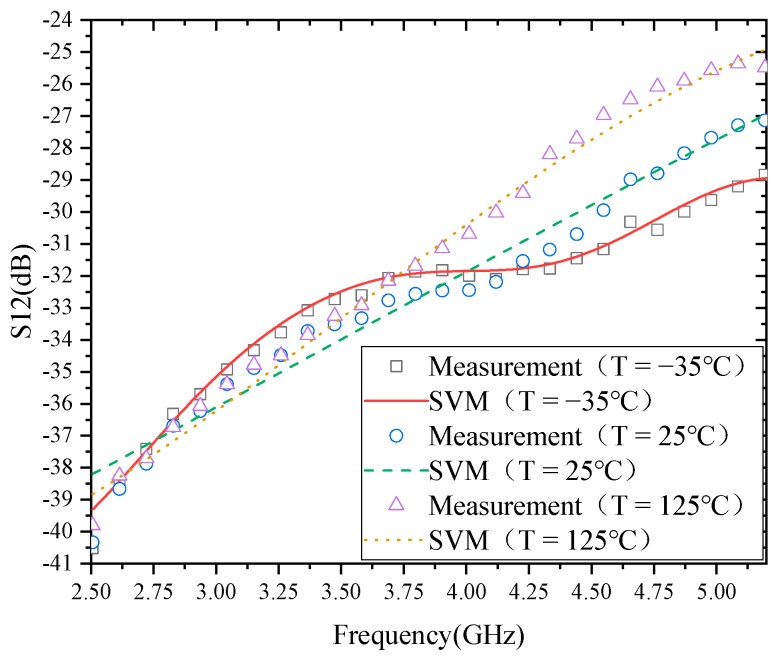
Modeling results of S12.

**Figure 6 micromachines-13-01012-f006:**
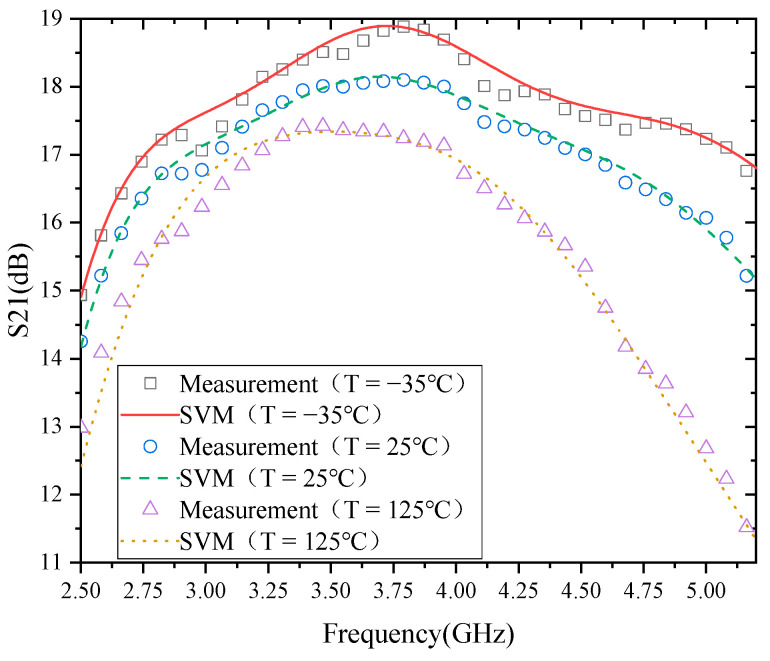
Modeling results of S21.

**Figure 7 micromachines-13-01012-f007:**
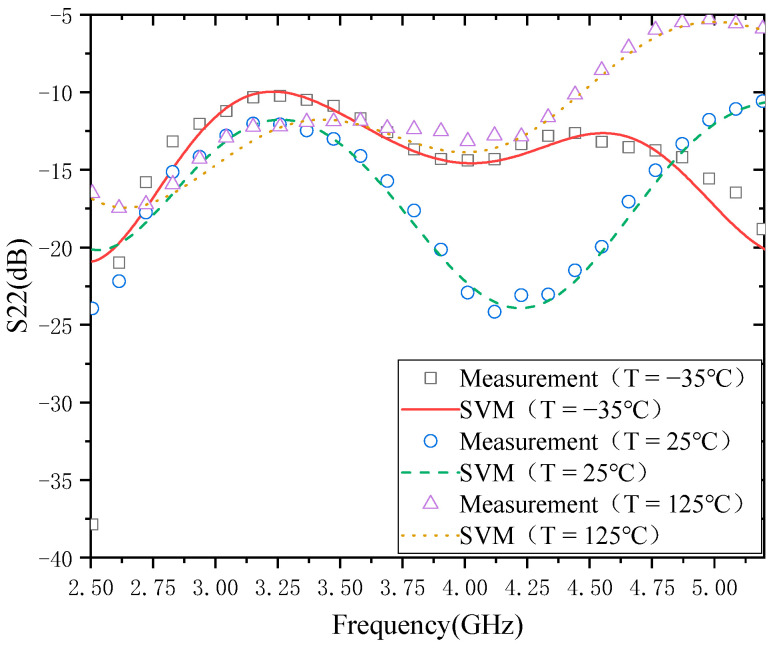
Modeling results of S22.

**Figure 8 micromachines-13-01012-f008:**
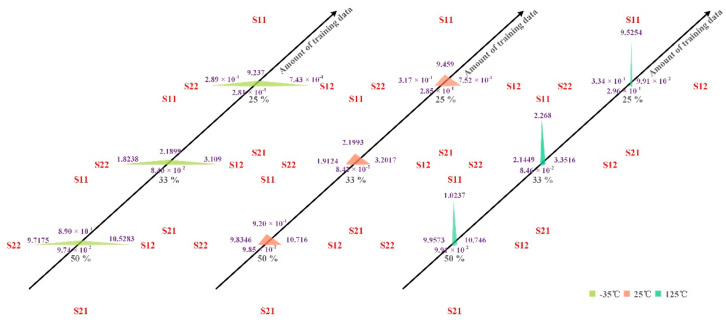
Training error of different amounts of training data on the model.

**Figure 9 micromachines-13-01012-f009:**
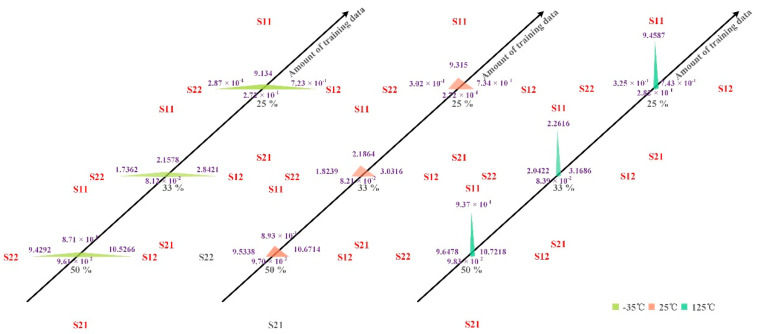
Test error of different amounts of training data on the model.

**Figure 10 micromachines-13-01012-f010:**
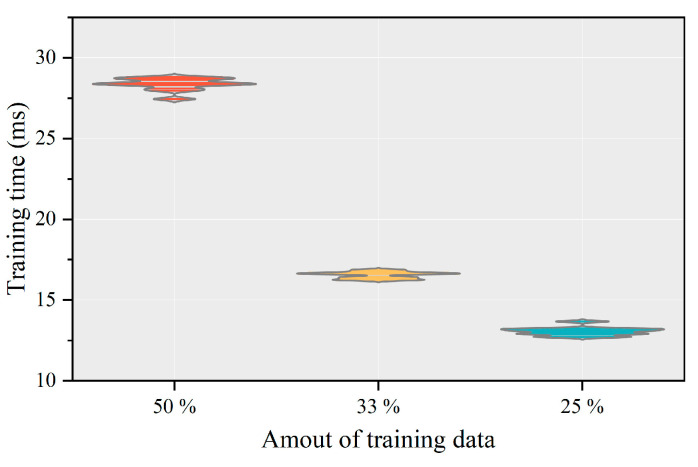
The training time of different amounts of training data on the model.

**Table 1 micromachines-13-01012-t001:** Effect of different amounts of training data on the model.

Specification	Temperature	Amount of Training Data (50%)		Amount of Training Data (33%)		Amount of Training Data (25%)	
		Training Error (MSE)	Test Error (MSE)	Training Error (MSE)	Test Error (MSE)	Training Error (MSE)	Test Error (MSE)
S11	−35 °C	8.9011 × 10^−1^	8.7065 × 10^−1^	9.197 × 10^−1^	8.926 × 10^−1^	1.0237	9.374 × 10^−1^
	25 °C	2.1899	2.1578	2.1993	2.1864	2.268	2.2616
	125 °C	9.237	9.134	9.459	9.315	9.5254	9.4587
S12	−35 °C	10.5283	10.5266	10.716	10.6714	10.746	10.7218
	25 °C	3.109	2.8421	3.2017	3.0316	3.3516	3.1686
	125 °C	7.4322 × 10^−1^	7.2301 × 10^−1^	7.5193 × 10^−1^	7.3363 × 10^−1^	7.6452 × 10^−1^	7.4296 × 10^−1^
S21	−35 °C	9.7368 × 10^−2^	9.6149 × 10^−2^	9.8468 × 10^−2^	9.7004 × 10^−2^	9.9113 × 10^−2^	9.8342 × 10^−2^
	25 °C	8.3957 × 10^−2^	8.1235 × 10^−2^	8.4159 × 10^−2^	8.2079 × 10^−2^	8.4565 × 10^−2^	8.3932 × 10^−2^
	125 °C	2.8103 × 10^−1^	2.719 × 10^−1^	2.8465 × 10^−1^	2.7202 × 10^−1^	2.9577 × 10^−1^	2.8169 × 10^−1^
S22	−35 °C	9.7175	9.4292	9.8346	9.5338	9.9573	9.6478
	25 °C	1.8238	1.7362	1.9124	1.8239	2.1449	2.0422
	125 °C	2.8863 × 10^−1^	2.8704 × 10^−1^	3.1704 × 10^−1^	3.0184 × 10^−1^	3.3387 × 10^−1^	3.2511 × 10^−1^

## Data Availability

Not applicable.

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
