# Peer review of "Support Vector Machine–Based Model for 2.5–5.2 GHz CMOS Power Amplifier"

_micromachines, 2022, doi:10.3390/mi13071012_

Round 1

Reviewer 1 Report

In this paper it is studied experimentally, using an Environment Chamber (Vötsch Industrietechnik SC3 1000 MHG which can provide an experimental environment of high and low temperatures from -40 ° C to 125 ° C), the modification of the specifications of a power amplifier (PA) in class A in 0.18μm CMOS technology under the action of temperature. To study the influence of changing PA specifications on the system, the authors used a vector support machine (SVM) to model the PA temperature characteristics. The experimental results showed that when the order of accuracy of the model is the same, the number of training data will be reduced by 25% and the training time of the model will be shortened by about 2.1 times.

Observation 1. The paper does not clearly mention what those specifications refer to, which change with the variation of temperature. A transistor in CMOS technology has mainly two parameters (as well as others indirectly influenced) that vary with temperature: one is a process parameter called the threshold voltage Vth and the other is μn (or μp) which is the carrier mobility. It is well known that these parameters have a coefficient of the temperature of order 1 and order 2. Is it about these parameters?

Observation 2. No schematic diagram of the power amplifier (PA) is also shown and it is not specified if there are coupling capacities that have a major influence on the SR parameter (supply regulator).

Observation 3. There are no calculation relations that show the theoretical way of modifying some parameters with the temperature and there is no correlation between the theoretical and the experimental part.

The paper is clearly experimental, well written and has important contributions. However, I consider it necessary to take into account the first two observations. I recommend the publication in the conditions of the modification of the paper.

Reviewer 2 Report

The abstract needs to be edited and the innovations of the article should be mentioned.

Circuit details of the PA used in this article is not provided. Will the accuracy and results of this work be a function of PA details as well?

In the flowchart of Figure 2, what is the maximum acceptable error and by what criteria is it selected?

Although it is said in this article, a model is presented, but in fact no model can be found in this article. What do the authors mean by the model?

It seems that the article needs to be fundamentally edited and its achievements should be compared with other similar methods.

Round 2

Reviewer 2 Report

  I think that the quality of figures can be improved.